

**Cooperative Effects of Field Traffic and Organic Matter Treatments on Some Compaction-Related Soil Properties**

Metin Mujdeci[1]*, Ahmet Ali Isildar[1], Veli Uygur[1], Pelin Alaboz[1], Husnu Unlu[2], Huseyin Senol[1]

[1]Suleyman Demirel University, Faculty of Agriculture, Department of Soil Science and Plant Nutrition, Isparta, Turkey.
[2]Suleyman Demirel University, Faculty of Agriculture, Department of Horticultural Sciences, Isparta, Turkey.

*Correspondence to*: Metin Mujdeci (*metinmujdeci@sdu.edu.tr*)

**Abstract.** Soil compaction is common problem of mineral soils under conventional tillage practices. Organic matter addition is an efficient way of reducing the effects of field traffic in soil compaction. The aim of this study was to investigate the effects of number of tractor passes (1, 3 and 5) on depth dependent (0-10 and 10-20 cm) penetration resistance, bulk density and porosity of a clay textured soil (Typic Xerofluvent) under organic vegetable cultivation practiced in 2010-2013 growing

seasons treated with farmyard manure (35 t FYM ha⁻¹), green manure (GM) (common vetch, *Vicia sativa* L.) and conventional tillage (C). The number of tractor passes resulted in increases in bulk density and penetration resistance (C>GM>FYM) whereas the volume of total and macro pores decreased. The maximum penetration resistance (3.60 MPa) was recorded in C treatment with 5 passes in 0-10 cm depth whereas the minimum (1.64 MPa) was observed for FYM treatment with 1 pass in 10-20 cm depth. The highest bulk density was determined as 1.61 g cm⁻³ for C treatment with 5

passes in 10-20 cm depth, the smallest value was 1.25 g cm⁻³ in the FYM treatment with only 1 pass in 0-10 cm depth. The highest total and macro pores volume were determined as 0.53 and 0.16 cm³ cm⁻³, respectively, in 0-10 cm depth of FYM treatment with 1 pass. The volume of micro pores (0.38 cm³ cm⁻³) was higher in 0-10 cm depth of FYM treatment with 3 passes. It can be concluded that organic pre-composted organic amendment rather than green manure was likely to be more efficient in chasing compaction problem in soils.


**Key Words:** soil penetration resistance, farmyard manure, green manure, bulk density, porosity

**1. Introduction**

Soil compaction is regarded as the most serious environmental problem resulting in conventional agriculture (McGarry, 2003). Since the farmers have difficulty in locating and rationalizing this type of degradation without making any measurement this problem can be more deleterious in conventional agriculture. In addition, compaction induced shallow plant rooting and poor plant growth reduce crop yield and vegetative cover to protect soil from erosion. Compaction can increase runoff and erosion from sloping land or waterlogged soils in flatter food slopes depending on reduced water

infiltration through soil surface (Al-Dousari et al., 2000; USDA-NRCS, 2012; Pulido et al., 2016). Intensive agricultural



related-soil compaction may be regarded as one of the significant reasons for land degradation (Cerda, 2000) and the elevated risks about food security, water scarcity, climate change, biodiversity loss and health threats which were pointed out as soil related chalanges for sustainable society (Keesstra et al., 2016).

The most significant cause of soil compaction, that can be defined as a soil degradation process in which an applied pressure
to a soil causes re-organisation to get closer of soil grains resulting in reduction in porosity and pore volume ratio, is field traffic. Meanwhile the close relation between field traffic density and frequency and plant type should also be taken into account (De Oliveira et al., 2015; Gelaw et al., 2015). Thus, more than 80% of a corn and soybean field has been under tyre pressure in a growth season (Erbach, 1986). 90%, 35% and 60% of field under wheel pressure during seed bed preparation, harvesting and baling practices, respectively, in cereal cultivation (Munsuz, 1985). Soil aeration, infiltration, and hydraulic
conductivity parameters of soils which are closely related to soil porosity show decreases related to increased-field traffic (Seker and Isildar, 2000; Aksakal, 2004). In order to prevent from such adverse effects, a decreased field traffic is necessary along with tillage soil at optimum moisture content and increasing the organic matter content of soil with farmyard manure, compost, green manure, etc. addition. Stabilization and fortification of soil aggregates with organic matter addition can increase compaction resistance of soils (Cochrane and Aylmore, 1994; Thomas et al., 1996; Aksakal et al., 2016), and
enhance to a differing degree the compaction related attributes such as bulk density, pore-size distribution, infiltration, etc. in soils (Sparovek et al., 1999; Carter, 2002; Aksakal et al., 2016). These differences are likely to be related to the nature of organic matter, C/N ratio and the degree of resistance to decomposition, soil type, and environmental conditions such as moisture and temperature (Hamza and Anderson, 2005). The changes in soil organic carbon stock under different management systems (Munoz et al., 2015) can also influence total soil quality in soils. For example, Parras-Alcantra et al.
(2015) reported a higher organic farming induced-stratification ratio deeper in the surface horizon comparing to conventional agricultural system. Gelaw et al. (2015) pointed out that the management of soil differently affected the partition of organic matter in different aggregate sizes which in turn influenced bulk density and water stable aggregates. Despite the specific effect of field traffic was not elucidated in these studies (Gelaw et al., 2015; Parras-Alcantra et al., 2015), the management systems are closely related to traffic density and soil physical attributes.

Keesstra et al. (2016) pointed out the significance of raising public/farmers' awareness about key attributes of soil organic matter to function and sustain ecosystem services. Similarly many researcher reported that soil physical and chemical properties in terms of fertility and sustainability of agriculture may be enhanced, to a large extent, by regular organic matter application (Aggelides and Londra, 2000; Alagoz et al., 2006; Mamman et al., 2007; Celik et al., 2010; Gulser and Candemir, 2012). The above mentioned literature points out that the nature and extent of compaction induced soil
degradation can be exaggerated by the lack of organic matter. Artificial loosening of soils by deep ripping is common suggested practice for elimination of the deleterious effects of compaction but its effect is not long-lasting. Therefore, the aim of this study was to investigate the effects of both different organic matter addition and field traffic density on penetration resistance, bulk density, porosity in a clay textured soil with low organic matter content.



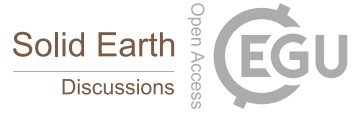

## 2. Materials and Methods

### Study area and experimental design

This study was carried out in experimental field of Agricultural Research and Application Centre of Suleyman Demirel University in 2010-2013 years. Some chemical and physical properties of soil were given in Table 1.

Organic vegetables were cultivated with farmyard manure (FYM), green manure (GM) and conventional tillage without any organic matter addition (C) treatments. The field experiment was set up in completely randomised design with three replications. FYM application was executed in between $21^{rd}$ of May and $7^{th}$ of June at 35 t ha$^{-1}$ (FYM consist of 45% dry matter) and then thoroughly mixed into surface layer (10-15 cm) of soils by rototiller. Common vetch (*Vicia sativa* L.) was sowed in the second week of March and allowed to grow till flowering stage, between $21^{rd}$ of May and $7^{th}$ of June then it was

incorporated into the soil by firstly chisel (20-25 cm tillage depth) and followed by rototillers (10-15 cm tillage depth). Both GM and FYM treatments were performed at the same time. The field was tilled by means of disk-harrow at 10-15 cm depth (16.09.2013) then the field was sprinkler-irrigated on $27^{th}$ of September 2013. In order to increase the susceptibility of soils to compaction, field traffic was realised at soil water content just below the field capacity (0.23 g g$^{-1}$) on $5^{th}$ of November 2013. A tractor (80-66s fiat, manufactured in 1998) having 85 horse power (HP) and 3460 kg weight, including the operator

weight, was used to resemble field traffic. Tractor passes (1, 3 and 5 times) were performed at 5 km h$^{-1}$ speed on the same track. One disturbed and three undisturbed soil cores per plot were taken from 0-10 and 10-20 cm depths before tractor passes and only undisturbed core samples after each passing treatment. Penetration resistance which is indication of soil compaction was measured with a penetrometre (Eijelkelkamp penetrograph) equipped with 1 cm$^2$ cone attachment. Penetration resistance were performed in 15 replications per treatment at 2 m interval and 0-10 and 10-20 cm depth. The

averages of resistance values obtained for 0-10 and 10-20 cm depth were evaluated. Some physical and chemical properties of experimental soil, FYM and plots before passing were given in Table 1 and 2. Particle size distribution was determined by means of Bouyoucos hydrometer method (Bouyoucos, 1962) and bulk density by undisturbed soil cores (Blake and Hartge, 1986). For this analysis soil cores were oven-dried and weighed then calculated by dividing the dry weight to core volume. Total porosity was determined in undisturbed water-saturated core samples of 100 cm$^3$. Micro porosity was accounted as the

volume of water at field capacity, measured by using a pressure membrane apparatus at 0.033 MPa suction. Macro porosity was calculated from the difference between total porosity and micro porosity (Danielson and Sutherland, 1986). Wet oxidation method of modified-Walkley and Black and dry ashing method at 400°C for 16 h in an oven were used for organic matter content of soil and FYM, respectively (Burt, 2004). Soil pH and electrical conductivity (EC) were measured in 1:1 soil: distilled water suspension whereas pH and EC of FYM were determined in 1:2.5 FYM: distilled water suspension (Burt,

2004). Carbonate equivalent was determined by a volumetric method using Scheibler calcimeter (Kacar, 2009).



**Statiscial analysis**

The data were subjected to ANOVA using MINITAB 16 statistical package programme (Minitab, 2010). The mean separation between the treatments was performed by LSD test at 95% confidence level.

Table 1. Some chemical and physical properties of experimental soil and FYM.

|  | Clay (g kg⁻¹) | Silt (g kg⁻¹) | Sand (g kg⁻¹) | Texture class | Organic matter (g kg⁻¹) | CaCO₃ (%) | pH | EC (dS m⁻¹) |
|---|---|---|---|---|---|---|---|---|
| Soil |  |  |  |  |  |  |  |  |
| 0-10 cm | 425.1 | 394.5 | 180.4 | C | 15.5 | 24.1 | 7.44 | 0.34 |
| 10-20 cm | 412.9 | 399.9 | 187.2 | C | 15.6 | 24.2 | 7.38 | 0.38 |
| FYM | - | - | - | - | 451 | - | 7.70 | 3.58 |

Table 2. Porosity, bulk density, organic matter content of the plots before passing.

| Treatments | Depth (cm) | Total porosity (cm³ cm⁻³) | Macro porosity (cm³ cm⁻³) | Micro porosity (cm³ cm⁻³) | Bulk density (g cm⁻³) | Organic matter (g kg⁻¹) |
|---|---|---|---|---|---|---|
| C | 0-10 | 0.54 | 0.22 | 0.32 | 1.28 | 15.50 |
|  | 10-20 | 0.52 | 0.19 | 0.33 | 1.33 | 15.60 |
| FYM | 0-10 | 0.61 | 0.24 | 0.37 | 1.14 | 28.00 |
|  | 10-20 | 0.58 | 0.22 | 0.36 | 1.22 | 21.50 |
| GM | 0-10 | 0.59 | 0.23 | 0.36 | 1.20 | 17.40 |
|  | 10-20 | 0.55 | 0.20 | 0.35 | 1.25 | 18.30 |

## 3. Results and Discussion

### 3.1 Soil penetration resistance

15 Organic matter amendments significantly reduced penetration resistance in both depths; however the effect of FYM treatment was higher than GM treatment (Fig. 1, Table 3). This finding is in accordance with the previous studies (Celik et al., 2010; Gulser and Candemir, 2012; Xin et al., 2016). Incorporation of organic matter to clay-textured soils can strenghten the aggregates by weakening cohesion forces and interfering with the formation of crust and large aggregate (Aksakal et al., 2012). The larger amounts of organic matter addition may mediate the formation of clay-organic matter complexes which in





fact reduces the penetration resistance on one hand and conserves organic matter against microbial decay on the other hand. In this respect, Blanco-Moure et al. (2016) investigated the effect of soil texture on carbon and organic matter distribution among different fractions under different tillage and management practices. They found that soil clay had critical role in chemical stabilization of organic matter through clay-organic complexes in the soils. Czyz and Dexter (2016) pointed out the

relation between the magnitude of clay-soil complex and porous and open nature of stucture. Thus, stable organic matter sources such as FYM resulted in desirable penetration resistance for plant growth under changing field traffic.

Table 3. Main effects of organic matter incorporation, depth and passing number on measured parameters.

| Treatments | Penetration resistance (MPa) | Bulk density (g cm$^{-3}$) | Total porosity (cm$^3$ cm$^{-3}$) | Micro porosity (cm$^3$ cm$^{-3}$) | Macro porosity (cm$^3$ cm$^{-3}$) |
|---|---|---|---|---|---|
| Organic matter | | | | | |
| C | 2.46 a | 1.53 a | 0.424 c | 0.336 c | 0.087 b |
| FYM | 2.02 c | 1.40 c | 0.471 a | 0.364 a | 0.107 a |
| GM | 2.27 b | 1.47 b | 0.445 b | 0.354 b | 0.091 b |
| Depth | | | | | |
| 0-10 cm | 2.44 a | 1.43 b | 0.458 a | 0.354 a | 0.104 a |
| 10-20 cm | 2.06 b | 1.50 a | 0.435 b | 0.349 b | 0.086 b |
| Passes | | | | | |
| 1 | 1.91 c | 1.38 c | 0.480 a | 0.349 b | 0.131 a |
| 3 | 2.10 b | 1.48 b | 0.441 b | 0.360 a | 0.082 b |
| 5 | 2.73 a | 1.54 a | 0.418 c | 0.345 c | 0.073 c |

Different letters in the same column indicates differences between the treatment means for each main effect.

Increasing number of passing irrespective from the organic matter treatments and soil depth increased penetration resistance

or soil compaction (Table 3). The effect of field traffic on penetration resistance, as expected, was more negative in depths near the surface (Table 3). Accordingly, Carman (1994) and Seker and Isildar (2000) were determined higher compaction ratio in the 0-10 and 0-15 cm surface layers, respectively. A penetration resistance value as high as 3.60 MPa in the surface layer caused by 5 times passing in control treatment with no organic matter may have significant inverse effects on infiltration, percolation, and runoff induced erosion under intensive precipitation events on slopy lands. In this study, we also

determined this well-known manner that surface soil become more compact with field traffic and the severity of the problem may be overcome by organic matter addition into soil. Penetration resistance value in 10-20 cm depths obtained for C and GM treatments after 5 passes was over 2 MPa value which considered limit value by USDA (1993) as a critical physical quality parameter in conventional agricultural practices. This critical value can change depending on the soil tillage systems, for example in minimum tillage practices where chisel is used for soil tillage it is 3 MPa and in no-till system it is 3.5 MPa

(De Moraes et al., 2014). Critical penetration resistance value inhibiting the root development is accepted as 3 MPa





(Busscher and Sojka, 1987; Hakansson and Lipiec, 2000, Aksakal et al., 2011). Soil management sytems can differ soil organic carbon contents (Munoz-Rojas et al.,2015) and field traffic density which ultimately degraded soil physical traits for optimal plant growth such as water stable aggregates and bulk density (Gelaw et al. 2015). In fact, these tendency in soil physical traits can lead a more compaction both in surface and subsurface  soil layer as in our case.

### 3.2. Bulk density

The main effect of organic matter treatments on bulk density was statistically significant (P<0.01). Both FYM and GM incorporation into soil were distinctly different than the control (Table 3). FYM amendment reduced the bulk density to as low as 1.40 g cm$^{-3}$. Similar to our findings, organic matter amendment-induced-decreases (Haynes and Naudi, 1998; Chaudhari et al., 2013; Gulser and Candemir, 2012) and field traffic related increases (Seker and Isildar, 2000; Patel and Mani, 2011) in bulk density have been frequently reported in the literature. The magnitude of above mentioned changes were depth dependent. In terms of plant growth, Aksakal et al. (2016) reported the enhancing effect of increasing vermicompost treatment rate on bulk density for three soils with differing clay content. In general, mixing of soil with less dense organic material results in decreased particle density in soils amended with organic manures (Haynes & Naidu, 1998). However, their efficiency to improve bulk density for plant growth is related to the magnitude and quality of organic residues (Aksakal et al., 2016). Green manure which is largely decomposed off by leaving relatively smaller extent of organic matter have limited influence on soil physical atrributes (Sauerbeck, 1982). Since FYM is more stable in terms of decomposition resistance than the GM, more organic compounds accumulated in soils (Table 2) treated with FYM. This infact mediated the formation of aggregates resistant to soil traffic and therefore minimum bulk density was obtained in FYM plots.  The main effect of soil depth irrespective to passing number and organic matter addition was significant (Table 3). Surface layer had smaller bulk density. The minimum bulk density (1.25 g cm$^{-3}$) was obtained in 0-10 cm depth for one pass in FYM treatments whereas the maximum one (1.61 g cm$^{-3}$) was recorded in 10-20 cm depth in C treatment with 5 passes (Fig. 2). The depth of soil compaction in soil profile is dependent on the axle load, soil moisture content, tire size, contact pressure, traffic density, and soil organic matter content related attributes such as aggregation, aggregate stability, porosity, etc. (Hamza and Anderson, 2005). The  greater axle loads and wetter soils consolidates the deeper in the soil profile. Since in our case these two dependents were taken constant, the effects of organic matter treatments are rather apparent. The stronger structures induced by larger amount and decomposition resistant organic substances scattered the force to a larger area which minimized the compaction-induced bulk density differences in FYM treated plots at any given pass number. At any steady state condition in terms of organic matter addition each treatment may be regarded as fixed condition but with differing passes number such as in our case for any treatment, the increasing number of traffic density resulted in increases in the bulk density deeper in the profile (Fig. 2). Parras-Alcantra et al. (2015) similarly reported that organic farming comparing to conventional tillage significantly improved soil organic carbon stocks of soil which resulted in a decrease of soil bulk density in the soil profile as deep as 76.1 cm. Vice versa, organic matter loss upon conversion of forest soil to ag-lands have made



soil progressively bulkier for vertisols and ultisols in 20 years (Bruun et al., 2015) which in turn explain the significance of organic matter to manage bulk density.

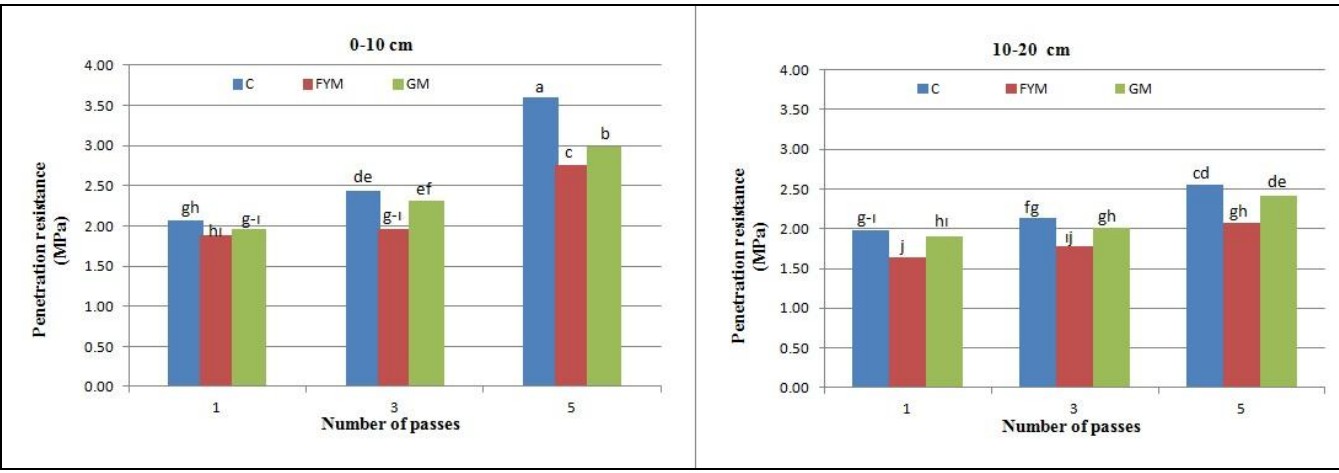

Figure 1. The effects number of passes on penetration resistance in different depths (passing×depth×treatment; P<0.01).
5   Different letters written above columns indicate the difference in the treatment means at P<0.05.

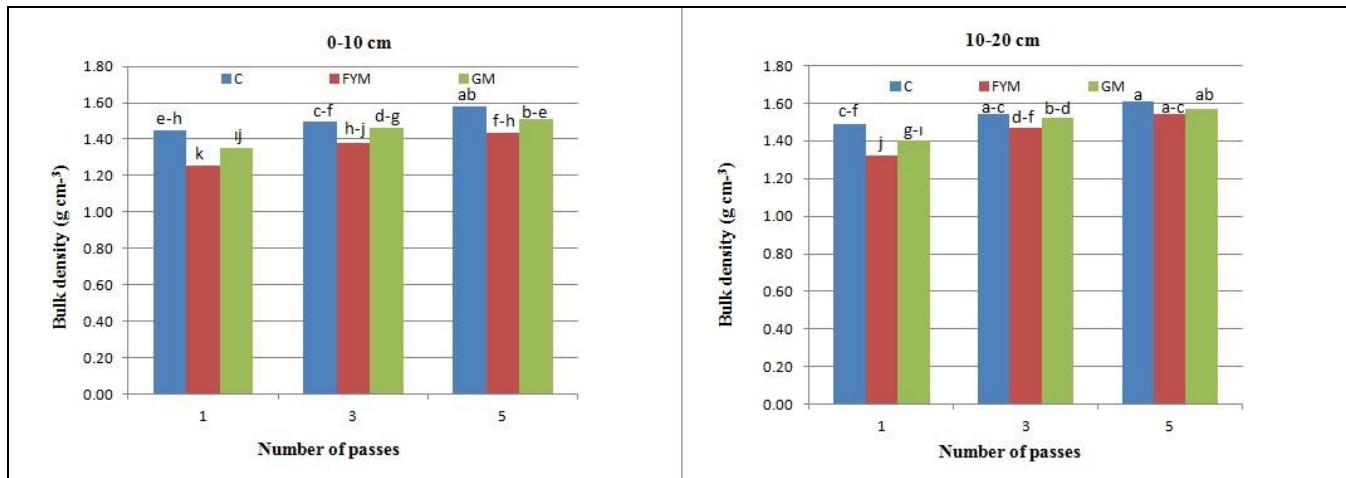

Figure 2. Number of passes and soil depth dependent bulk density changes induced by organic matter treatments (passing×treatment, depth×treatment; P<0.01). Different letters written above columns indicate the difference in the
10   treatment means at P<0.05.



### 3.3. Porosity

The main effects of organic matter incorporation, depth and passes number on porosity were significant (Table 3). The effects of treatments in relation to soil depth and field traffic density on porosity in descending order were FYM>GM>C. The main overall effect of depth on total porosity was detrimental in 10-20 cm depth where a significant decrease was observed. The initial average (0.565 cm$^3$ cm$^{-3}$) of total porosity calculated from Table 2 showed a detrimental decrease down to 0.418 and 0.441 cm$^3$ cm$^{-3}$ after 5 and 3 passes (Table 3). As all plots considered, the maximum and minimum values of total porosity were 0.53 and 0.39 cm$^3$ cm$^{-3}$ respectively (Fig 3). The effects of organic matter treatments, depth and field traffic density on micro porosity was statistically significant (Table 3). Microporosity and total porosity parameters similarly responded to organic matter treatments. The volume of micropores were significantly higher in surface layer than the one observed in 10-20 cm (Table 3). Maximum micro porosity (0.38 cm$^3$ cm$^{-3}$) was recorded in FYM treatment in 0-10 cm depth with 3 passes whereas the minimum (0.32 cm$^3$ cm$^{-3}$) was determined in C treatment in the surface layer (0-10 cm) with 5 passes. There was an increase in the micro porosity of control plot upon one pass comparing to its corresponding initial porosity, in contrast a decrease was recorded for FYM and GM plots. As three passes were done, it increased in all treatments including the control. Initial micro porosity value and the one after 5 passes were nearly the same in C treatment but they dropped below even the initial porosity for FYM and GM treatments. Despite the organic matter incorporation was made only four years the enhancement was recorded about porosity parameters. The enhancement induced by organic matter amendments in long term studies at various locations was even more astonishing and show similar manner to our findings. For example, Rasool et al. (2008) and Arthur et al. (2013) reported increases in total porosity and water retention which is related to pore nature of soil with increase in organic matter content depending on agricultural practices or organic matter amendments. The FYM and GM, to relatively smaller extent, promotes total and micro porosity in the current study. Organic matter application to soils promotes the development of better soil structure by binding the soil particles with polysaccharides and bacterial exudates which results in decreased bulk density and hence the porosity (Bhatia and Shukla, 1982). As the level of soil compaction increased, the amount of water held in high matric potentials decreases whereas the magnitude of water held at low matric potentials is to increase (Gupta et al., 1989) due to convertion of some macropores into micropores by compression stress. Similarly, Seker and Isildar (2000) reported an increase in the pore volume holding plant available water after 4 passes. The descending order of the treatments was FYM>GM>C for both of the depths and each pass-treatment and as pointed out by Celik et al. (2004) micro porosity increased upon organic matter amendments.

Macro porosity which is critical for soil aeration and soil water circulation were changed as a function of soil depth and field traffic density and organic mattter amendments (Table 3). The main effect of organic matter was in descending order as FYM>GM>C. In this study, organic matter amendments significantly improved macro porosity (P<0.01). However, field traffic at 3 and 5 passes (Table 3) reduced the macropores volume below the critical level of 0.1 cm$^3$ cm$^{-3}$ (Hakansson and Lipiec, 2000) and covered on the conditioner effects of organic amendments. The maximum macro porosity (0.16 cm$^3$ cm$^{-3}$)





was observed in 0-10 cm depth of FYM treatment at one pass whereas the minimum was recorded for C treatment in 10-20 cm depth at 5 passes (Fig 5).

Enhancement in soil structure traits by reduction of aggregate wettability (Zhang and Hartge, 1992) strength of aggregate stability with organic matter incorporation partially eliminated the effects of field traffic on macro porosity after four

consecutive years of FYM and GM application. The organic matter bound-ambiguity was attributed to type of organic matter, C/N ratio and the degree of resistance to decomposition. Readily decomposable soil organic matter reported to be more relevant than total organic matter in mechanical characterization of the soil (Ball et al., 2000). For example the less humified organic matters, such as green manure, was reported to highly efficient to increase aggregate porosity (Zhang, 1994) but this effects was found to be not durable and resistant to field traffic as compared to the effects of FYM in the

current study. Since, the overall effect of GM to increase soil organic matter are much lesser than the one maintained by FYM which more stable and resistant to microbial decay such behaviour is likely in organic matter poor soils.

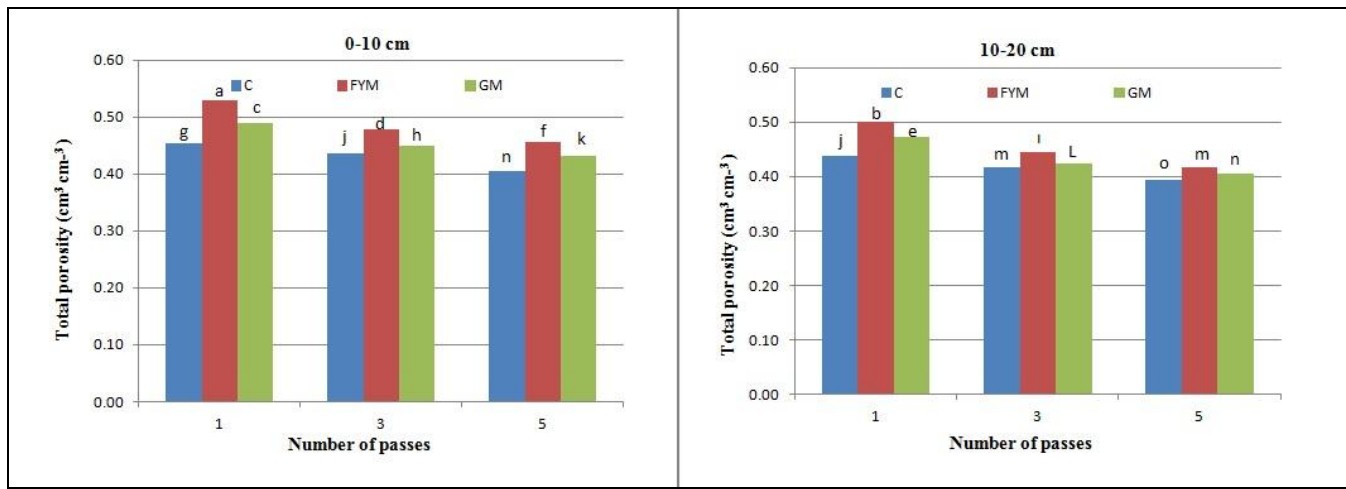

Figure 3. Number of passes and soil depth dependent total porosity changes induced by organic matter treatments

(passing×depth×treatment; P<0.01). Different letters written above columns indicate the difference in the treatment means at P<0.05.





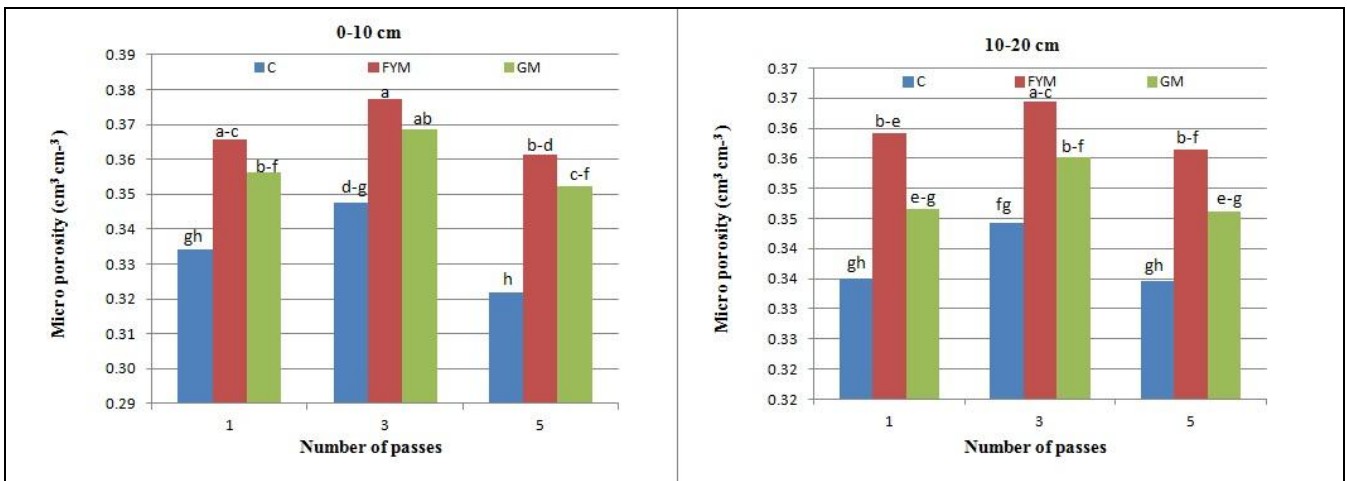

Figure 4. Number of passes and soil depth dependent micro porosity changes induced by organic matter treatments (passing×depth; P<0.05, depth×treathment; P<0.01) Different letters written above columns indicate the difference in the treatment means at P<0.05.

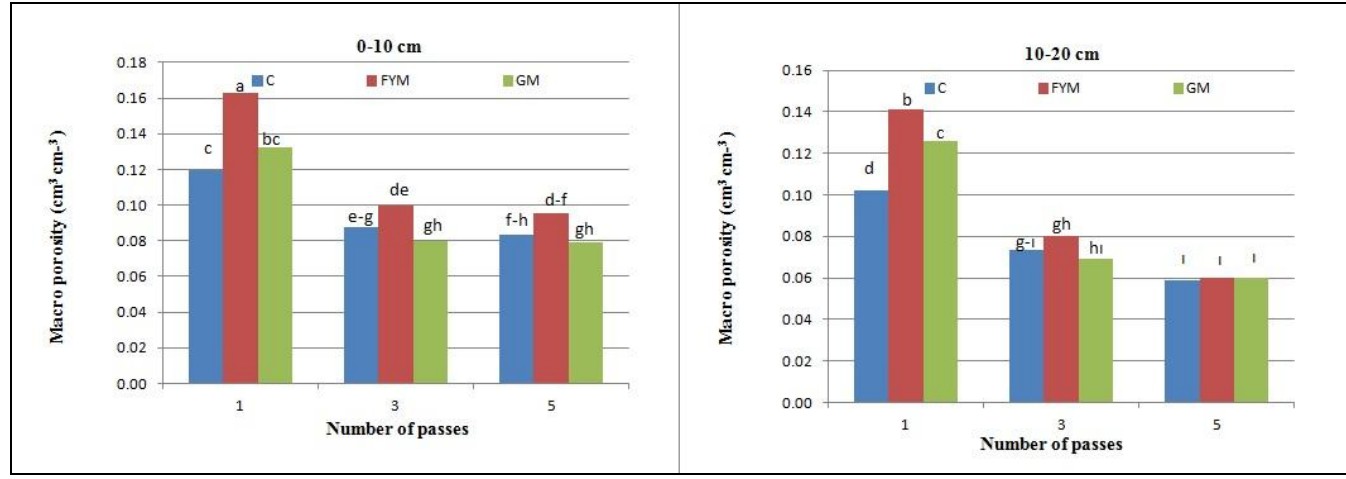

Figure 5. Number of passes and soil depth dependent macro porosity changes induced by organic matter treatments (passing× depth, depth×treathment, passing×treathment; P<0.01). Different letters written above columns indicate the
10   difference in the treatment means at P<0.05.

## 4. Conclusions

15   The effect of field traffic density on soil compaction was found to be dependent on type of organic matter treatment. The

mentioned differences were rather related to amounts of organic matter treatment than type of organic matter. The overall





effects of organic treatments irrespective from soil depth on penetration resistance and bulk density were in descending order as C>GM>FYM whereas it was FYM>GM>C for total and micro porosity. Macro porosity appeared to be higher at minimum field traffic for FYM treatment in the surface layer. It can be concluded that the use of organic matter enhance soil conditions by influencing the soil water holding and circulation characteristics, aeration, penetration resistance and bulk
density which have implications on plant root growth.

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
