# Peer review of "Cooperative Effects of Field Traffic and Organic Matter Treatments on Some Compaction-Related Soil Properties"

_Solid Earth, 2016_

## Referee Comment (RC1) · Anonymous Referee #1 · 9 Aug 2016

Dear author I found your paper of interest but you need to improve the graphs The columns should show different brown colors from light to dark (from 1 to 5 passes) and remove the frame and the parallel lines. Please make unique graphs (you are using the template of Excel) your paper needs to be stronger with the citations Please, See my comments Sincerely

Please also note the supplement to this comment:
http://www.solid-earth-discuss.net/se-2016-84/se-2016-84-RC1-supplement.pdf

[Figure]

**Supplement:**

[revised manuscript text omitted]

---

## Short Comment (SC1) · 10 Aug 2016

This paper is study on effect of organic materai application on the soil compaction. The paper deals with on interesting aspect and present a wide dataset. This study is still able to add some knowlodge of interest to an environmental readerships. Iconcluded that this paper is suited for the original paper.

---

## Referee Comment (RC2) · Anonymous Referee #2 · 5 Sep 2016

The paper entitled "Cooperative Effects of Field Traffic and Organic Matter Treatments on Some Compaction-Related Soil Properties" deals with a very classical subject about soil compaction in a agricultural land in Turkey. Paper is interesting, but it is not performed to be understood. Authors deals with well-known problem. Methods and conclusions are not new neither revolutionary. In paper missing data for better understanding the methods, very important for understanding the results. There is no data related to crop performance, or for stability of aggregates, although authors explain great part of their results through state of structural stability in treatments. Missing explanations of the methods used in the research. What volume had cylinders for bulk density? Why authors did not measure (or estimate?) quantity of common vetch incorporated

into soil? In how many occasions organic amendments were incorporated into soil? The paper has no novelty/originality and I wonder how this paper can contribute to the community of readership? Some of conclusions are speculations and they are not supported with results. Specific comments are below. Page – line Page 1- line 6-7 – It is conclusion statement and should go on the end of paragraph. Page 1 – line 19 – try to use another word (e.g. mitigation). I have doubts that "Chasing" was the best choice. Page 1 – line 25 – Please define soil compaction before authors started the introduction. Page 1 – line 25-26 – Soil compaction occurs also trough natural processes as well, not only as consequence of anthropogenic activity Page 1 – line 27-28 – It depends about depth of compacted soil layer. There are crops that do not need deeper rhizosphere than 20 cm. Page 2 – line 4-6 – Definition of compaction goes at the beginning of Introduction section. Page 2 –line 6 – Please use term crop instead plant. Do that in whole manuscript. Page 2 – line 7-9 - Try to use newer literature sources, if possible. Page 2 – line 10 - put "differential porosity" instead "soil porosity". Page 2 – line 11-13 – Sentence is unclear. Please reformulate. Page 2 – line 16 – What differences? Please define. Page 2 – line 17-18 – Authors should add more reference on this statement. This is very broad and attractive subject so please add more references if authors speak about international level. Page 2 – line 31 – Please add reference that supports this statement. This is common research on most of different textured soil types. Page 2 – line 33 – Authors missed to write hypothesis and specific objectives of paper. Explanation of paper novelty should also be valuable to the readers. Page 3 – line 7-11 – Did authors applied farmyard manure and green manure each year? If so please state that. Page 3 – line 16-17 - Please state a number of soil cores in total. Page 3 – line 19. Unclear. Penetrometer (Ejkelkamp) record soil strength at each centimetre. Page 3 – line 24 – authors should provide more detailed procedure for determination total porosity. Page 4 - Line 3-4 – Statistical analysis is not described well. Readers cannot follow used procedures about post-hoc test. Do your datasets follow normal distribution? Page 4 - Line 15 – Significance in not confirmed at all number of passes (Fig. 1) so this statement is incorrect. Page 5 – line 6 – Define "desirable

penetration resistance" Page 5 – line 9 –Please write: . . .. increased soil compaction measured by penetration resistance. Page 5 –line 10-11 – define which depths. Page 5 – line 12 - Put source (Fig 1) and support the written information Page 5 – line 12-14 – This is speculation. Authors did not measure those parameters. I suggest putting reference that supports this statement. Page 5 – line 16 – Add: . . ..or by adoption soil management system with decreased annual traffic. Page 6 – line 28-29 – It will be of great benefit for the paper quality if authors could provide structural stability test. There are numerous of simple handled tests in literature. Page 6 – line 35 – define ag-lands Page 7 – please remove borders around figures. Also remove horizontal lines. Also state did significance was determined at 0.01 or 0.05 level? Page 8 – line 11-13 – Please add Fig number that supports this information. Page 8 –line 16-17 – It is very hard to follow discussion when it is not clear when organic amendment is incorporated in soil? 2010? 2013? Or each year? Page 10 –line 15 – Please add: ". . . to be dependent to addition and type of organic matter treatment." Page 10 – line 15-16 – This is speculation. Where is author's data that support your statement? Authors did not state amount of green manure incorporated into soil.

---

## Author Comment (AC1) · 4 Oct 2016

The introduction section of the manuscript was improved according to the suggestions of the referee one and the changes made were incorporated into the text and written in red.

Please also note the supplement to this comment:
http://www.solid-earth-discuss.net/se-2016-84/se-2016-84-AC1-supplement.pdf
* * *
[Figure]

[Figure]

**Fig. 1.**

[Figure]

**Fig. 2.**

[Figure]

**Fig. 3.**

[Figure]

**Fig. 4.**

[Figure]

**Fig. 5.**

**Supplement:**

[revised manuscript text omitted]

Barbero-Sierra, C., Marques, M.J., Ruiz-Pérez, M., Escadafal, R., Exbrayat, W.: How is Desertification Research Addressed in Spain? Land Versus Soil Approaches. Land Degrad. Develop., 26 (5), 423-432, 2015.

Bhatia, K.S. and Shukla, K.K.: Effect of continuous application of fertilizers and manure on some physical properties of eroded alluvial soil, J. Indian Soc. Soil Sci., 30, 33–36, 1982.

30 Blake, G.R. and Hartge, K.H.: Bulk density, in: Methods of Soil Analysis, Part 1, Physical and Mineralogical Methods, (Ed: A. Klute), Agr. Monogr. 9, ASA and SSSA, Madison WI., USA, 363-375, 1986.

Blanco-Moure, N., Gracia, R., Bielsa, A. C., and López M. V.: Soil organic matter fractions as affected by tillage and soil texture under semiarid Mediterranean conditions. Soil Tillage Res., 155 (Special Issue), 381-389, 2016.

Bouyoucos, G.J.: Hydrometer method improved for making particle size analyses of soils. Agron. J., 54, 464-465, 1962.

Bruun, T. B., Elberling, B., de Neergaard, A., and Magid, J.: Organic carbon dynamics in different soil types after conversion of forest to agriculture, Land Degrad. Develop., 26 (3), 272-283, 2015.

Burt, R. (Ed.): Soil Survey Laboratory Methods Manual, Soil Survey Investigations Rep. 42, version 4.0, USDA–NRCS, 2004.

Busscher, W.J. and Sojka, R.E.: Enhancement of subsoiling effect on soil strength by conservation tillage, Transac. Am. Soc. Agric. Engin., 30, 4, 888-892, 1987.

Carman, K.: Tractor forward volocity and tine load effects on soil strength, J. Terramech., 31, 11-20, 1994.

Carter, M.R.: Soil quality for sustainable land management: organic matter and aggregation interactions that maintain soil functions, Agron. J., 94, 38–47, 2002.

Celik, I., Gunal, H., Budak, M., and Akpinar C.: Effects of long-term organic and mineral fertilizers on bulk density and penetration resistance in semi-arid Mediterranean soil conditions, Geoderma, 160, 236-243, 2010.

Celik, I., Ortas, I., and Kilic, S.: Effects of compost, mycorrhiza, manure and fertilizer on some physical properties of a Chromoxerert soil. Soil Tillage Res., 78, 59–67, 2004.

Cerdà A.: Aggregate stability against water forces under different climates on agriculture land and scrubland in southern Bolivia. Soil Tillage Res., 57, 159–166, 2000.

Chaudhari, P.R., V. Ahire, D.V., Ahire, V.D., Chkravarty, M., and Maity, S.: Soil bulk density as related to soil texture, organic matter content and available total nutrients of Coimbatore soil. Int. J. Sci. Res. Pub., 3(2), 1-8, 2013.

Cochrane, H.R. and Aylmore, L.A.G.: The effects of plant roots on soil structure, in: Proceedings of 3rd Triennial Conference ''Soils 94'', 207–212, 1994.

Czyz, E.A. and Dexter, A.R.: Estimation of the density of the clay-organic complexes in soil. Int. Agrophys., 30 (1), 19-23, 2016.

Danielson, R.E. and Sutherland, P.L.: Porosity, in: Methods of Soil Analysis, Part 1, Physical and Mineralogical Methods. 2nd ed. (Ed: A. Klute) Agron. Monogr. 9. ASA and SSSA, Madison WI., USA, 443-461, 1986.

De Moraes, M. T., Debiasi, H., Carlesso, R., Franchini, J.C., and Da Silva, V.R.: Critical limits of soil penetration resistance in a rhodic eutrudox, R. Bras. Ci. Solo, 38, 288-298, 2014.

De Oliveira, S. P., de Lacerda, N. B., Blum, S. C., Escobar, M. E. O., and de Oliveira, T. S.: Organic carbon and nitrogen stocks in soils of northeastern Brazil converted to irrigated agriculture, Land Degrad. Develop., 26 (1), 9-21, 2015.

Erbach, D.: Farm equipment and soil compaction. ASAE Paper No: 860730, 1986.

Gelaw, A. M., Singh, B. R., Lal R.: Organic carbon and nitrogen associated with soil aggregates and particle sizes under different land uses in Tigray, Northern Ethiopia, Land Degrad. Develop., 26 (7), 690-700, 2015.

Gulser, C. and Candemir, F.: Changes in penetration resistance of a clay field with organic waste applications. Eurasian J. Soil Sci., 1, 16–21, 2012.

Gupta, S.C., Sharma, P.P., and DeFranchi, S.A.; Compaction effects on soil structure, in: Advances in Agronomy, Ed. N.C. Brady, 42, 311-338, 1989.

Hakansson, I. and Lipiec, J.: A review of the usefulness of relative bulk density values in studies of soil structure and compaction. Soil Tillage Res., 53, 71-85, 2000.

5    Hamza, M.A. and Anderson, W.K.: Responses soil properties and grain yields to deep ripping and gypsum application in a compacted loamy sand soil contrasted with a sandy clay loam soil in Western Australia. Austuralian Journal of Agricultural Res. 54, 273-282, 2003.

Hamza, M.A. and Anderson, W.K.: Soil compaction in cropping systems: a review of the nature, causes and possible solutions. Soil Tillage Res., 82, 121–145, 2005.

10    Hamza, M.A. and Anderson, W.K.: Combinations of ripping depth and tine spacing for compacted sandy and clayey soils. Soil & Tillage Res., 99,  213–220, 2008.

Haynes, R.J. and Naidu, R.: Influence of lime, fertilizer and manure applications on soil organic matter content and soil physical conditions: a review. Nutr. Cyc. Agroecosys., 51, 123–137, 1998.

15    Kacar, B.: Soil Analysis, Nobel publications, Ankara (in Turkish), 2009.

Kaleeem Abbasi M., Mahmood Tahir M., Sabir N., Khurshid M. Impact of the addition of different plant residues on nitrogen mineralization-immobilization turnover and carbon content of a soil incubated under laboratory conditions.  Solid Earth, 6 (1), 197-205, 2015.

Keesstra, S.D., Bouma, J., Wallinga, J., Tittonell, P., Smith, P., Cerdà, A., Montanarella, L., Quinton, J.N., Pachepsky, Y.,
20    van der Putten, W.H.,  Bardgett, R.D., Moolenaar, S., Mol, G., Jansen, B., and Fresco, L.O.: The significance of soils and soil science towards realization of the United Nations Sustainable Development Goals. Soil, 2, 111–128, 2016.

Kozlowski, T.T.:  Soil Compaction and Growth of Woody Plants. Scand. J. For. Res. 14, 596-619, 1999.

Mamman, E., Ohu, J. O., and Crowther, T.: Effect of soil compaction and organic matter on the early growth of maize (Zea mays) in a vertisol, Int. Agrophys., 21, 367-375, 2007.

25    McGarry, D.: Tillage and soil compaction, in: Conservation Agriculture, L. Garcia-Torres et al. (eds.), 307-316, 2003.

Minitab.: Minitab 16 Statistical Software, Minitab Inc., State College, Pennsylvania, USA, 2010.

Muñoz-Rojas, M., Jordán, A., Zavala, L. M., De la Rosa D., Abd-Elmabod, S. K., and Anaya-Romero,  M.: Impact of land use and land cover changes on organic carbon stocks in mediterranean soils (1956-2007). Land Degrad. Develop., 26 (2), 168-179, 2015.

30    Munsuz, N.: Soil Mechanics and Technology, Publications of Ankara Univ. Faculty of Agriculture, 922, Text book, 260, Ankara (in Turkish),  1985.

Ni, J., Luo, D.H., Xia, J., Zhang, Z.H., and Hu, G.: Vegetation in karst terrain of southwestern China allocates more biomass to roots. Solid Earth, 6 (3), 799-810, 2015.

Novara A., Gristina L., Guaitoli F., Santoro A., Cerdà A. Managing soil nitrate with cover crops and buffer strips in Sicilian vineyards. Solid Earth, 4 (2), 255-262, 2013.

Ola, A., Dodd, I.C., and Quinton, J.N.: Can we manipulate root system architecture to control soil erosion? Soil 1, 603-612, 2015.

5 Parras-Alcántara, L., Díaz-Jaimes, L., and Lozano-García, B.: Organic farming affects C and N in soils under olive groves in mediterranean areas. Land Degrad. Develop., 26 (8), 800-806, 2015.

Patel, S.K. and Mani, I.: Effect of multiple passes of tractor with varying normal load on subsoil compaction. J. Terramech., 48, 277–284, 2011.

Peng F., Quangang Y., Xue X., Guo J., Wang T. Effects of rodent-induced land degradation on ecosystem carbon fluxes in
10 an alpine meadow in the Qinghai-Tibet Plateau, China. Solid Earth, 6 (1), 303-310, 2015.

Pulido, M., Schnabel, S., Contador, J. F. L., Lozano-Parra, J., and González, F.: The impact of heavy grazing on soil quality and pasture production in rangelands of SW Spain. Land Degrad. Develop., DOI: 10.1002/ldr.2501, 2016.

Rasool, R., Kukal, S. S., and Hira, G. S.: Soil organic carbon and physical properties as affected by long-term application of FYM and inorganic fertilizers in maize-wheat system, Soil Tillage Res., 101 (1-2), 31-36, 2008.

15 Sauerbeck, D.R.: Influence of crop rotation, manurial treatment and soil tillage on the organic matter content of German soils. In: Soil Degradation, Boels D, Davies DB & Johnston AE (eds), Proceedings of the EEC Seminar held in Wageningen, Rotterdam, A A Balkema, Netherlands, pp 163–179, 1982.

Seker, C. and Isildar, A.A.: Effects of wheel traffic porosity and compaction of soil profile. Turk. J. Agric. Forest., 24, 71–77, 2000.

20 Shaw, E.A., Denef, K., Milano de Tomasel, C., Cotrufo, M.F., and Wall, D.H.: Fire affects root decomposition, soil food web structure, and carbon flow in tallgrass prairie. Soil, 2, 199-210, 2016.

Sparovek, G., Lambais, M.R., Silva, A.P., and Tormena, C.A.: Earthworm (Pontoscolex corethrurus) and organic matter effects on the reclamation of an eroded oxisol. Pedobiologia, 43, 698–704, 1999.

Thomas, G.W., Haszler, G.R., and Blevins, R.I.: The effect of organic matter and tillage on maximum compactibility of soils
25 using the proctor test. Soil Sci., 161, 502–508, 1996.

USDA, Soil Survey Manual, Soil Survey Division Staff, Washington, DC, USA, 1993.

USDA-NRCS: Soil Compaction. http://www.nrcs.usda.gov/Internet/FSE_DOCUMENTS/stelprdb1187272.pdf. 27.05.2016, 2012.

Xin, X., Zhang, J., Zhu, A., and Zhang, C.: Effects of long-term (23 years) mineral fertilizer and compost application on
30 physical properties of fluvo-aquic soil in the North China Plain. Soil Tillage Res., 156, 166–172, 2016.

Wang, T., Xue, X., Zhou, L., Guo, J.: Combating Aeolian Desertification in Northern China. Land Degrad. Develop., 26 (2), 118-132, 2015.

Yan, X., and Cai, Y.L.: Multi-Scale Anthropogenic Driving Forces of Karst Rocky Desertification in Southwest China. Land Degrad. Develop., 26 (2), 193-200, 2015.

Zhang, H. and Hartge, K.H.: Effect of differently humified organic matter on the aggregate stability by reducing aggregate wettability. Z. Pflanzenernahr. Bodenkd., 155, 143–149, 1992.

Zhang, H.: Organic matter incorporation affects mechanical properties of soil aggregates. Soil Tillage Res., 31, 263–275, 1994.

---

## Author Comment (AC2) · 4 Oct 2016

The changes made in light of the second referee's suggestions were written in green. Some specific responses were also given below.

Page – line Page 1- line 6-7 –It is conclusion statement and should go on the end of paragraph.

R: Despite it can be regarded as a concluding sentences this sentence describes the common problem and possible solution to the problem. Therefore we think that the sentence may be used as the rational as well. Page 1 – line 19 –try to use another word (e.g. mitigation R: mitigation was replaced for chasing Page 1 – line 25 – Please

define soil compaction before authors started the introduction. R: The definition of the compaction was given at the start after the rational.

Page 1 – line 27-28 – It depends about depth of compacted soil layer. There are crops that do not need deeper rhizosphere than 20 cm.

R: the case was specified for deep rooting plants

Page 2 – line 4-6 – Definition of compaction goes at the beginning of Introduction section.

R: It moved to the introductory part.

Page 2 –line 6 – Please use term crop instead plant. Do that in whole manuscript.

R: The appropriate ones were replaced.

Page 2 – line 10 - put "differential porosity" instead "soil porosity". R: it was done. Page 2 – line 17-18 – Authors should add more reference on this statement. This is very broad and attractive subject so please add more references if authors speak about international level.

R: New references were added.

Page 2 – line 31 – Please add reference that supports this statement. This is common research on most of different textured soil types

R: References were added.

Page 3 – line 16-17 - Please state a number of soil cores in total. R: Three soil cores were taken from each plot fro each tractor passeses.

Page 6 – line 28-29 – It will be of great benefit for the paper quality if authors could provide structural stability test.

R: Unfortunately we are not able to present the agragate stability data of soils. Beacause the analysis was not performed in 2013.

Page 6 – line 35 – define ag-lands R: agricultural lands

Page 7 – please remove borders around figures. Also remove horizontal lines. Also state did significance was determined at 0.01 or 0.05 level?

R: The frame and horizontal lines werre removed from the Figs. The significance level used was 0.05.

Page 10 – line 15-16 – This is speculation. Where is author's data that support your statement? Authors did not state amount of green manure incorporated into soil.

R: The sentence was omitted in order to avoid from the speculation.

Minor language corrections in the light of referee's suggestions were written in green and therefore specific responses to those comments were not given.

Please also note the supplement to this comment:
http://www.solid-earth-discuss.net/se-2016-84/se-2016-84-AC2-supplement.pdf
* * *
[Figure]

**Fig. 1.**

[Figure]

[Figure]

**Fig. 2.**

[Figure]

**Fig. 3.**

[Figure]

**Fig. 4.**

[Figure]

[Figure]

**Fig. 5.**

**Supplement:**

[revised manuscript text omitted]

Barbero-Sierra, C., Marques, M.J., Ruiz-Pérez, M., Escadafal, R., Exbrayat, W.: How is Desertification Research Addressed in Spain? Land Versus Soil Approaches. Land Degrad. Develop., 26 (5), 423-432, 2015.

Bhatia, K.S. and Shukla, K.K.: Effect of continuous application of fertilizers and manure on some physical properties of eroded alluvial soil, J. Indian Soc. Soil Sci., 30, 33–36, 1982.

30 Blake, G.R. and Hartge, K.H.: Bulk density, in: Methods of Soil Analysis, Part 1, Physical and Mineralogical Methods, (Ed: A. Klute), Agr. Monogr. 9, ASA and SSSA, Madison WI., USA, 363-375, 1986.

Blanco-Moure, N., Gracia, R., Bielsa, A. C., and López M. V.: Soil organic matter fractions as affected by tillage and soil texture under semiarid Mediterranean conditions. Soil Tillage Res., 155 (Special Issue), 381-389, 2016.

Bouyoucos, G.J.: Hydrometer method improved for making particle size analyses of soils. Agron. J., 54, 464-465, 1962.

Bruun, T. B., Elberling, B., de Neergaard, A., and Magid, J.: Organic carbon dynamics in different soil types after conversion of forest to agriculture, Land Degrad. Develop., 26 (3), 272-283, 2015.

Burt, R. (Ed.): Soil Survey Laboratory Methods Manual, Soil Survey Investigations Rep. 42, version 4.0, USDA–NRCS, 2004.

Busscher, W.J. and Sojka, R.E.: Enhancement of subsoiling effect on soil strength by conservation tillage, Transac. Am. Soc. Agric. Engin., 30, 4, 888-892, 1987.

Carman, K.: Tractor forward volocity and tine load effects on soil strength, J. Terramech., 31, 11-20, 1994.

Carter, M.R.: Soil quality for sustainable land management: organic matter and aggregation interactions that maintain soil functions, Agron. J., 94, 38–47, 2002.

Celik, I., Gunal, H., Budak, M., and Akpinar C.: Effects of long-term organic and mineral fertilizers on bulk density and penetration resistance in semi-arid Mediterranean soil conditions, Geoderma, 160, 236-243, 2010.

Celik, I., Ortas, I., and Kilic, S.: Effects of compost, mycorrhiza, manure and fertilizer on some physical properties of a Chromoxerert soil. Soil Tillage Res., 78, 59–67, 2004.

Cerdà A.: Aggregate stability against water forces under different climates on agriculture land and scrubland in southern Bolivia. Soil Tillage Res., 57, 159–166, 2000.

Chaudhari, P.R., V. Ahire, D.V., Ahire, V.D., Chkravarty, M., and Maity, S.: Soil bulk density as related to soil texture, organic matter content and available total nutrients of Coimbatore soil. Int. J. Sci. Res. Pub., 3(2), 1-8, 2013.

Cochrane, H.R. and Aylmore, L.A.G.: The effects of plant roots on soil structure, in: Proceedings of 3rd Triennial Conference ''Soils 94'', 207–212, 1994.

Czyz, E.A. and Dexter, A.R.: Estimation of the density of the clay-organic complexes in soil. Int. Agrophys., 30 (1), 19-23, 2016.

Danielson, R.E. and Sutherland, P.L.: Porosity, in: Methods of Soil Analysis, Part 1, Physical and Mineralogical Methods. 2nd ed. (Ed: A. Klute) Agron. Monogr. 9. ASA and SSSA, Madison WI., USA, 443-461, 1986.

De Moraes, M. T., Debiasi, H., Carlesso, R., Franchini, J.C., and Da Silva, V.R.: Critical limits of soil penetration resistance in a rhodic eutrudox, R. Bras. Ci. Solo, 38, 288-298, 2014.

De Oliveira, S. P., de Lacerda, N. B., Blum, S. C., Escobar, M. E. O., and de Oliveira, T. S.: Organic carbon and nitrogen stocks in soils of northeastern Brazil converted to irrigated agriculture, Land Degrad. Develop., 26 (1), 9-21, 2015.

Erbach, D.: Farm equipment and soil compaction. ASAE Paper No: 860730, 1986.

Gelaw, A. M., Singh, B. R., Lal R.: Organic carbon and nitrogen associated with soil aggregates and particle sizes under different land uses in Tigray, Northern Ethiopia, Land Degrad. Develop., 26 (7), 690-700, 2015.

Gulser, C. and Candemir, F.: Changes in penetration resistance of a clay field with organic waste applications. Eurasian J. Soil Sci., 1, 16–21, 2012.

Gupta, S.C., Sharma, P.P., and DeFranchi, S.A.; Compaction effects on soil structure, in: Advances in Agronomy, Ed. N.C. Brady, 42, 311-338, 1989.

Hakansson, I. and Lipiec, J.: A review of the usefulness of relative bulk density values in studies of soil structure and compaction. Soil Tillage Res., 53, 71-85, 2000.

5    Hamza, M.A. and Anderson, W.K.: Responses soil properties and grain yields to deep ripping and gypsum application in a compacted loamy sand soil contrasted with a sandy clay loam soil in Western Australia. Austuralian Journal of Agricultural Res. 54, 273-282, 2003.

Hamza, M.A. and Anderson, W.K.: Soil compaction in cropping systems: a review of the nature, causes and possible solutions. Soil Tillage Res., 82, 121–145, 2005.

10    Hamza, M.A. and Anderson, W.K.: Combinations of ripping depth and tine spacing for compacted sandy and clayey soils. Soil & Tillage Res., 99,  213–220, 2008.

Haynes, R.J. and Naidu, R.: Influence of lime, fertilizer and manure applications on soil organic matter content and soil physical conditions: a review. Nutr. Cyc. Agroecosys., 51, 123–137, 1998.

15    Kacar, B.: Soil Analysis, Nobel publications, Ankara (in Turkish), 2009.

Kaleeem Abbasi M., Mahmood Tahir M., Sabir N., Khurshid M. Impact of the addition of different plant residues on nitrogen mineralization-immobilization turnover and carbon content of a soil incubated under laboratory conditions.  Solid Earth, 6 (1), 197-205, 2015.

Keesstra, S.D., Bouma, J., Wallinga, J., Tittonell, P., Smith, P., Cerdà, A., Montanarella, L., Quinton, J.N., Pachepsky, Y.,
20    van der Putten, W.H.,  Bardgett, R.D., Moolenaar, S., Mol, G., Jansen, B., and Fresco, L.O.: The significance of soils and soil science towards realization of the United Nations Sustainable Development Goals. Soil, 2, 111–128, 2016.

Kozlowski, T.T.:  Soil Compaction and Growth of Woody Plants. Scand. J. For. Res. 14, 596-619, 1999.

Mamman, E., Ohu, J. O., and Crowther, T.: Effect of soil compaction and organic matter on the early growth of maize (Zea mays) in a vertisol, Int. Agrophys., 21, 367-375, 2007.

25    McGarry, D.: Tillage and soil compaction, in: Conservation Agriculture, L. Garcia-Torres et al. (eds.), 307-316, 2003.

Minitab.: Minitab 16 Statistical Software, Minitab Inc., State College, Pennsylvania, USA, 2010.

Muñoz-Rojas, M., Jordán, A., Zavala, L. M., De la Rosa D., Abd-Elmabod, S. K., and Anaya-Romero,  M.: Impact of land use and land cover changes on organic carbon stocks in mediterranean soils (1956-2007). Land Degrad. Develop., 26 (2), 168-179, 2015.

30    Munsuz, N.: Soil Mechanics and Technology, Publications of Ankara Univ. Faculty of Agriculture, 922, Text book, 260, Ankara (in Turkish),  1985.

Ni, J., Luo, D.H., Xia, J., Zhang, Z.H., and Hu, G.: Vegetation in karst terrain of southwestern China allocates more biomass to roots. Solid Earth, 6 (3), 799-810, 2015.

Novara A., Gristina L., Guaitoli F., Santoro A., Cerdà A. Managing soil nitrate with cover crops and buffer strips in Sicilian vineyards. Solid Earth, 4 (2), 255-262, 2013.

Ola, A., Dodd, I.C., and Quinton, J.N.: Can we manipulate root system architecture to control soil erosion? Soil 1, 603-612, 2015.

5 Parras-Alcántara, L., Díaz-Jaimes, L., and Lozano-García, B.: Organic farming affects C and N in soils under olive groves in mediterranean areas. Land Degrad. Develop., 26 (8), 800-806, 2015.

Patel, S.K. and Mani, I.: Effect of multiple passes of tractor with varying normal load on subsoil compaction. J. Terramech., 48, 277–284, 2011.

Peng F., Quangang Y., Xue X., Guo J., Wang T. Effects of rodent-induced land degradation on ecosystem carbon fluxes in
10 an alpine meadow in the Qinghai-Tibet Plateau, China. Solid Earth, 6 (1), 303-310, 2015.

Pulido, M., Schnabel, S., Contador, J. F. L., Lozano-Parra, J., and González, F.: The impact of heavy grazing on soil quality and pasture production in rangelands of SW Spain. Land Degrad. Develop., DOI: 10.1002/ldr.2501, 2016.

Rasool, R., Kukal, S. S., and Hira, G. S.: Soil organic carbon and physical properties as affected by long-term application of FYM and inorganic fertilizers in maize-wheat system, Soil Tillage Res., 101 (1-2), 31-36, 2008.

15 Sauerbeck, D.R.: Influence of crop rotation, manurial treatment and soil tillage on the organic matter content of German soils. In: Soil Degradation, Boels D, Davies DB & Johnston AE (eds), Proceedings of the EEC Seminar held in Wageningen, Rotterdam, A A Balkema, Netherlands, pp 163–179, 1982.

Seker, C. and Isildar, A.A.: Effects of wheel traffic porosity and compaction of soil profile. Turk. J. Agric. Forest., 24, 71–77, 2000.

20 Shaw, E.A., Denef, K., Milano de Tomasel, C., Cotrufo, M.F., and Wall, D.H.: Fire affects root decomposition, soil food web structure, and carbon flow in tallgrass prairie. Soil, 2, 199-210, 2016.

Sparovek, G., Lambais, M.R., Silva, A.P., and Tormena, C.A.: Earthworm (Pontoscolex corethrurus) and organic matter effects on the reclamation of an eroded oxisol. Pedobiologia, 43, 698–704, 1999.

Thomas, G.W., Haszler, G.R., and Blevins, R.I.: The effect of organic matter and tillage on maximum compactibility of soils
25 using the proctor test. Soil Sci., 161, 502–508, 1996.

USDA, Soil Survey Manual, Soil Survey Division Staff, Washington, DC, USA, 1993.

USDA-NRCS: Soil Compaction. http://www.nrcs.usda.gov/Internet/FSE_DOCUMENTS/stelprdb1187272.pdf. 27.05.2016, 2012.

Xin, X., Zhang, J., Zhu, A., and Zhang, C.: Effects of long-term (23 years) mineral fertilizer and compost application on
30 physical properties of fluvo-aquic soil in the North China Plain. Soil Tillage Res., 156, 166–172, 2016.

Wang, T., Xue, X., Zhou, L., Guo, J.: Combating Aeolian Desertification in Northern China. Land Degrad. Develop., 26 (2), 118-132, 2015.

Yan, X., and Cai, Y.L.: Multi-Scale Anthropogenic Driving Forces of Karst Rocky Desertification in Southwest China. Land Degrad. Develop., 26 (2), 193-200, 2015.

Zhang, H. and Hartge, K.H.: Effect of differently humified organic matter on the aggregate stability by reducing aggregate wettability. Z. Pflanzenernahr. Bodenkd., 155, 143–149, 1992.

Zhang, H.: Organic matter incorporation affects mechanical properties of soil aggregates. Soil Tillage Res., 31, 263–275, 1994.